# Dynamic of Phenolic Compounds, Antioxidant Activity, and Yield of Rhubarb under Chemical, Organic and Biological Fertilization

**DOI:** 10.3390/plants9030355

**Published:** 2020-03-11

**Authors:** Alexandru Cojocaru, Laurian Vlase, Neculai Munteanu, Teodor Stan, Gabriel Ciprian Teliban, Marian Burducea, Vasile Stoleru

**Affiliations:** 1Department of Horticultural Technologies, “Ion Ionescu de la Brad” University of Agricultural Sciences and Veterinary Medicine, 3 M. Sadoveanu, 700440 Iasi, Romania; cojocaru.alexandru@yahoo.com (A.C.); nmunte@uaiasi.ro (N.M.); steodor@uaiasi.ro (T.S.); gabrielteliban@uaiasi.ro (G.C.T.); marian.burducea@yahoo.com (M.B.); 2Department of Bio-pharmaceutics and Pharmaceutical Technology, Iuliu Hatieganu University of Medicine and Pharmacy, 41 Victor Babes, 400012 Cluj Napoca, Romania; laurian.vlase@umfcluj.ro; 3“Alexandru Ioan Cuza” University of Iasi, 11 Bd. Carol, 700506 Iasi, Romania

**Keywords:** *Rheum rhabarbarum* L., fertilization, yield, polyphenols, DPPH^+^

## Abstract

In recent years, rhubarb is being increasingly cultivated, as it provides early yields when the vegetables supply to market is deficient and shows high levels of both polyphenols content and antioxidant capacity in edible parts. In 2017, we investigated crops of the rhubarb cultivar Victoria to the fifth year of production. Comparisons were performed between three root phase fertilizations—chemical (NPK 16-16-16^®^), organic (Orgevit^®^), and biological (Micoseeds MB^®^)—plus an unfertilized control. The determinations of polyphenols, the antioxidant capacity, and the yield indicators from the stalks (petioles) of rhubarb were made at each out of the 10 harvests carried out. The highest yield (59.16 t·ha^−1^) was recorded under the chemical fertilization. The total polyphenols content and antioxidant capacity varied widely from 533.86 mg GAE·g^−1^ d.w. and 136.86 mmol Trolox·g^−1^ d.w., respectively in the unfertilized control at the last harvest, up to 3966.56 mg GAE·g^−1^ d.w. and 1953.97 mmol Trolox·g^−1^ d.w. respectively under the organic fertilization at the four harvest. From the results of our investigation, it can be inferred that the chemical fertilization was the most effective in terms of yield, whereas the sustainable nutritional management based on organic fertilizer supply led to higher antioxidant compounds and activity.

## 1. Introduction

Rhubarb (*Rheum rhabarbarum* L.) is a perennial plant [1] that is characterized by high antioxidant properties and is cultivated only for its petiole [2], as its leaves are toxic due to their content of oxalic acid [3]. In Europe, it is cultivated mainly in Germany, France, and England [4]. Stalks are used for the preparation of various culinary preparations (soup, pie, compote, sweetness), but they are also used in traditional medicine as laxatives, gastrointestinal hemorrhage, and the treatment of constipation jaundice and ulcer [5]. A study has shown promising anti-cancer properties and the broad therapeutic potential of anthraquinines in the petioles of rhubarb [6]. The consumption of rhubarb in large quantities has an adverse effect on the accumulation of calcium in the body [3].

The technology and the nutritional regime used to cultivate crops influence the petioles production and its composition [7]. Important benefits from crop production have been observed by applying radicular fertilizers especially when soil conditions are limiting root uptake [8]. This way, much lower quantities of fertilizers are required to sustain growth [9]. Due to the global population growth and the need to provide food, chemical fertilization remains the most used. Detailed studies have shown that chemically fertilized crops show yields varying widely from 26.39 t·ha^−1^ to 39.08 t·ha^−1^ [10]. Due to the growth of organic products in recent years, farmers have also shifted to less polluting fertilization means, such as organic ones, which provide economically satisfactory yields [11]. At the same time, the food safety of the fresh or processed product is a necessity [12].

Fertilizers and harvesting influence the rhubarb phenolic content and composition as well as the antioxidant activity [13,14,15]. Health beneficial, antioxidative, antimicrobial, and plant protective properties are influenced by the synergistic effect of polyphenolic compounds [16,17,18]. In previous studies on rhubarb, total polyphenol content and composition as well as antioxidant activity were significantly affected by harvest dates, fertilizations, and the combined effect of these two factors [7,19]. Literature reports indicate variations in antioxidant capacity from 491 µmol Trolox·g^−1^ d.w., in case of the clone Victoria 574/27, up to 1242 µmol Trolox·g^−1^ d.w., in the case of the Valentin variety [5].

It should be emphasized that literature data regarding the effect of fertilizers (chemical and biological) as well as of the harvesting times on the polyphenol content of the rhubarb stalks are missing. Kalisz et al. analyzed the polyphenol content of two cultivars (Victoria and Red Malinowy) in the spring and autumn seasons, recording the highest value of flavan-3-ols for Red Malinowy cultivar from spring harvest (195.98 mg·g^−1^ d.w.) and the lowest for Victoria cv from autumn harvest (86.57 mg·g^−1^ d.w.) [13]. Studies on other vegetable species belonging to the family Polygonaceae were performed on *Rumex acetosa* [19,20,21,22], *Rumex scutatus* [23], *Rumex crispus* [24,25], *Rumex japonicus* [26], *Rumex hastatus* [27,28,29], *Rumex ecklonianus* [30], *Rumex tingitanus* [31], *Rumex sanguineus* [32], *Rumex acetosella* [33], *Rumex maderensis* [34], and *Rumex obtusifolius* [35], but no determinations relevant to polyphenols content and composition were performed.

The main objectives of this research were to assess: (1) the influence of the chemical, organic, and biological fertilization on the dynamic yield and their quality (polyphenols, antioxidant capacity, etc.); and (2) the optimal harvesting periods, correspondent to the highest plant nutrient content, in order to promote fertilization measures for sustainable agriculture.

## 2. Results and Discussion

The total polyphenol content and antioxidant capacity were positively influenced by both the type of fertilization applied and the date of harvest compared to the unfertilized control. For the total polyphenol content, the highest statistically significant increase was recorded in the fertilized treatment with Mo (61.68% compared to Ct). The organic fertilization (Og) provided 1477.95 mg GAE·g^−1^ d.w., followed by chemical fertilization with a total polyphenol content of 1320.62 mg GAE·g^−1^ d.w. (Table 1). The role of arbuscular mychorrhizal fungi AMF should be taken into consideration in the Mo treatment as an abiotic stress according to environmental conditions [36].

The highest polyphenol content was recorded at the fourth harvesting period (R_4_), with a value of 2450.86 mg GAE·g^−1^ d.w., followed by fifth harvest (R_5_) with a total polyphenol content of 1819.34 mg GAE·g^−1^ d.w. The lowest polyphenol content was recorded at the last harvest (R_10_), i.e., 670.92 mg GAE·g^−1^ d.w. Statistically, lower values were recorded at the eighth harvest time (R_8_) with a total polyphenol content of 1015.14 mg GAE·g^−1^ d.w. 

The content of TP increased from R_1_ to R_4_ and R_5_, after which it decreased toward the end of the production period, due to the reduction of the metabolism of the resent plants. Similar studies were also conducted in salad [37], where the total phenol content (TPC) was higher but dependent on the climatic conditions.

Regarding the influence of fertilization on antioxidant activity (AC), the highest values were recorded when fertilizing with Mo, with the value of 877.07 mmol Trolox·g^−1^ d.w. (280.06% against C), followed by the Og treatment, with the value of 728.05 mmol Trolox·g^−1^ d.w. (232.47% compared to C). Ch fertilization resulted in an antioxidant activity of 478.48 mmol Trolox·g^−1^ d.w. (152.52% compared to variant C).

Regarding the influence of the harvesting season on the antioxidant activity, the values ranged in the wide interval from 293.09 mmol Trolox·g^−1^ d.w., in the case of R_10_, up to 1079.76 mmol Trolox·g^−1^ d.w. in the case of the fourth harvesting season (R_4_). Higher statistically significant values of AC compared to C were recorded at R_5_ (834.44 mmol Trolox·g^−1^ d.w.) and at R_2_ (589.75 mmol Trolox·g^−1^ d.w.). Statistically lower values of AC compared to C were recorded at R_8_ (467.39 mmol Trolox·g^−1^ d.w.) and R_9_ (477.34 mmol Trolox·g^−1^ d.w.).

From the data presented in Table 1, a positive correlation was observed between the values of the total polyphenols content and the antioxidant capacity. Higher levels of antioxidant activity were recorded in the middle of the harvest period (R_4_ and R_5_).

Data obtained from this study are in agreement with those reported for other *Polygonaceae* familly species such as English spinach [38] and sorrel [39].

Regarding the influence of fertilization and harvesting time on the total polyphenols content, statistically significant differences were recorded (Table 2). 

Indeed, the total polyphenols broadly varied from 433.86 mg GAE·g^−1^ d.w., in the case of R_10_, unfertilized, up to 3966.56 mg GAE·g^−1^ d.w., in the case of R_10_, Og fertilization. (Table 2). In the case of Ch, the total polyphenols content ranged from 829.51 mg GAE·g^−1^ d.w., at R_8_, to 1738.26 mg GAE·g^−1^ d.w., in the case of R_4_. When fertilizing with Mo, the total polyphenol content influenced by the harvesting time ranged in a wide interval from 660.88 mg GAE·g^−1^ d.w., in case C, up to 2954.52 mg GAE·g^−1^ d.w., in case R_5_. Upon Og fertilization, the polyphenol content ranged from 387.72 mg GAE·g^−1^ d.w., in the case of R_6_, to 3966.56 mg GAE·g^−1^ d.w., in the case of R_4_. The unfertilized control showed the lowest polyphenols content at R_10_ (433.86 mg GAE·g^−1^ d.w.), while at R_4_, the highest value was recorded (2183.38 mg GAE·g^−1^ d.w.).

The data obtained in this experiment are in agreement with those obtained by Takeoka et al. who analyzed the content of polyphenols in the petioles of 29 different species of the genus *Rheum* harvested in two different years of production (2007 and 2009). The total polyphenols content ranged from 673 mg GAE·g^−1^ d.w., in the case of Loher Blut of the *Rheum officinale* species harvested in 2007, up to 4173 mg GAE·g^−1^ d.w., in the case of the Plum Hutt cv of the *Rheum rhabarbarum* L. species, harvested in 2009 [5]. Higher polyphenol content can be attributed to the effect of mycorrhizal associations, which induce a decrease in carbohydrate content in cells [40].

Regarding the influence of the fertilizer on the antioxidant capacity, this varied in wide limits from 136.86 mmol Trolox·g^−1^ d.w., in the case of C, from R_10_, to 1953.97 mmol Trolox·g^−1^ d.w., in the case of Og fertilization, from R_4_ (Table 3). 

Regarding the influence of Ch on harvesting times, the antioxidant capacity ranged from 300.55 mmol Trolox·g^−1^ d.w., in the case of R_8_, to 629.80 mmol Trolox·g^−1^ d.w., in the case of R_4_. Under Mo fertilization, the lowest values of AC were obtained at R_10_ (361.14 mmol Trolox·g^−1^ d.w.), while the highest values were recorded at R_5_ (1614.49 mmol Trolox·g^−1^ d.w.). As far as Og is concerned, the antioxidant activity depended on harvesting time and varied widely from 338.78 mmol Trolox·g^−1^ d.w., in the case of R_6_, to 1953.97 mmol Trolox·g^−1^ d.w., in case R_4_. Then, the lowest values at the last harvest (136.86 mmol Trolox·g^−1^ d.w.) and the highest ones were obtained at R_4_ (688.76 mmol Trolox·g^−1^ d.w.).

Solfanelli et al. justify the higher content of polyphenolic compounds by the fact that in warmer periods, with less precipitation, the sugar content increases in plants and thus the activity of AMF is higher [41].

Studies on the influence of different fertilizers on the antioxidant capacity of the petioles of rhubarb are scant. Zhou et al. determined the antioxidant activity of commonly consumed vegetables in Colorado and found significant variation among individual samples of each vegetable tested. They ranked the antioxidant activity of the vegetables as follows: rhubarb > green bean; tomato > potato; kale > spinach > broccoli [4].

Takeoka et al. analyzed the antioxidant capacity of the 29 species of the genus *Rheum*. The antioxidant capacity ranged from 463 µmol Trolox·g^−1^ d.w., in the case of *Rheum officinale*, up to 1242 µmol Trolox·g^−1^ d.w., in the case of Valentine cv, of the species *Rheum rhabarbarum* L. [5].

In perennial species, as in *Populus* sp., high TPC and AC are also accounted for by water stress and higher temperatures [42].

In general, polyphenols are secondary metabolism compounds that are produced by plants for defensive purposes under increased biotic or abiotic stress.

The phenolic compounds content and composition were positively influenced by the application of fertilizers and the harvesting season, and in this respect, p-coumaric acid, ferulic acid, isoquercitrin, rutoside, and quercetrol were analyzed (Table 4).

The values of p-coumaric ranged from 9.86 µg·g^−1^, in the case of Ch, to 15.46 µg·g^−1^, in the case of the Mo fertilization. Control and Og treatment had higher values of p-coumaric acid, of 17.44% and 11.86%, respectively, compared to the Ch treatment.

Ferulic acid widely varied from 21.67 µg·g^−1^, in the case of Ch, to 31.45 µg·g^−1^, in the case of the Mo treatment. Control and Og treatments showed increases of 24.87%, respectively 2.58%, compared to Ch.

Isoquercitrin was significantly influenced (*p* ≤ 0.05) by biological fertilization. Values ranged from 37.04 µg·g^−1^, in the case of Ch, to 78.65 µg·g^−1^, in the case of Mo. Control and Og fertilization resulted in increases of 84.12%, respectively 66.90%, compared to the Ch treatment.

Out of the phenolic compounds, rutozide showed the highest content. Rutoside ranged in a wide interval from 372.57 µg·g^−1^, in case C, to 490.97 µg·g^−1^, in the case of Og. Under fertilization with Ch and Mo, 27.01% increases were obtained, respectively 25.58%, compared to the unfertilized control.

Quercetrol varied from 11.57 µg·g^−1^ in the control to 20.20 µg·g^−1^ in the case of the fertilization with Mo. In the case of Og and Ch fertilization, 48.16% and 42.70% increases were obtained, compared to C.

The data from Table 4 regarding the influence of the treatment on the phenols quality highlight the favorable effect of the biofertilizers application on the crop, compared with Ch and C.

The harvesting time significantly influenced (*p* ≤ 0.05) phenol content as follows: the highest p-coumaric acid content was obtained at R_7_, ferulic acid at R_8_, isoquercitrin at R_6_, rutozide at R_4_, and quercetrol in R_9_. The lowest content of p-coumaric acid was recorded at first harvest (R_1_), while the lowest content of ferulic acid was recorded at R_5_, the lowest content of isoquercitrin and rutoside were recorded at the last harvest (R_10_), and the lowest content of quercetrol was recorded at R_9_.

P-coumaric acid varied within wide limits from 6.43 µg·g^−1^ in the case of R_1_ to 23.57 µg·g^−1^ in the case of R_7_. Statistically higher values of the p-coumaric acid content in the petioles of rhubarb were obtained also at the R_4_ and R_9_ harvest times, with increases of 286.96% and 268.3% respectively compared to R_1_, which indicates that this acid accumulates when the plants have overcome the early development stages, being also influenced by the climatic conditions [43].

Regarding the content of ferulic acid in the petioles of resin influenced by the harvesting season, it ranged from 15.69 µg·g^−1^ in the case of R_10_, to 39.96 µg·g^−1^, in the case of R_8_. Remarkable results were also obtained at the R_7_ and R_6_ times, with increases of 244.99% and 199.9% respectively compared to R_10_. Ferulic acid showed statistically low values in the case of R_9_ and R_8_, with increases of 4.90% and 6.8% respectively compared to R_10_. These results indicate that ferulic acid also accumulates in the second half of the harvesting period.

The values of isoquercitrin from the petioles of resin influenced by the harvesting time varied widely from 32.43 µg·g^−1^, in the case of R_10_, up to 84.83 µg·g^−1^, in the case of R_6_, which suggests that isoquercitrin accumulates in the middle of the harvesting period. Increases of 247.33% and 244.96% were obtained at R_9_ and R_8_, compared to R_10_. The low values of isoquercitrin were recorded at R_1_ and R_2_, with increases of 45.7% and 50% respectively compared to R_10_.

The content of rutoside in the petioles of resale varied in wide limits of 196.02 µg·g^−1^, in the case of R_10_, up to 904.12 µg·g^−1^, in the case of R_4_. A higher content of rutoside was obtained by R_5_ and R_2_, with increases of 340.1% and 254% respectively compared to the 10th harvest (R_10_). Lower values of rutoside were achieved by R_8_ and R_9_, with increases of 43.54% and 59.4% respectively compared to R_10_.

The content of quercetrol in the rhubarb petioles ranged from 11.22 µg·g^−1^, in the case of R_1_, to 21.64 µg·g^−1^, in the case of R_9_. Higher values were obtained by R_10_ and R_4_, with increases of 90.5% and 88% respectively compared to R_1_. Quercetrol achieved increases of 1.42% and 4.54% respectively in the R_2_ and R_3_ harvest periods.

Fertilization with Mo had a significant positive influence on p-coumaric acid, ferulic, isoquercitin, and quercetrol.

The positive influence of biological fertilizers on the content of polyphenols has also been demonstrated in species such as tomatoes [44,45] or peppers [46,47,48].

The type of fertilization and the harvesting season significantly influenced the content of the five polyphenols (p-coumaric acid, ferulic acid, isoquercitrin, rutoside, and quercetrol) (Table 5).

P-coumaric acid content widely varied from 4.00 µg·g^−1^, in the case of the Ch treatment, from the eighth period (R_8_), to 46.14 µg·g^−1^, in the case of fertilization with Mo from R_7_.

Ferulic acid fluctuated from 12.16 µg·g^−1^, in the case of Ch fertilization from R_5_, to 51.58 µg·g^−1^, in the case of Mo from R_7_.

In addition, isoquercitrin was also significantly influenced by the fertilizer and harvesting period. It ranged in wide limits from 25.50 µg·g^−1^, in the case of the Og fertilization, from R_6_, to 148.78 µg·g^−1^, in the case of the control, which was also from R_6_.

The values of rutoside fluctuated from 121.06 µg·g^−1^ in the case of the control from R_7_ to 1442.24 µg·g^−1^ in the case of Og fertilization from the R_4_ harvest time.

The content of quercetrol ranged from 6.78 µg·g^−1^ in the case of the control of R_7_ to 33.20 µg·g^−1^ in the case of the Ch treatment of R_7_ and Og from R_10_.

The content of biologically active compounds depends mainly on the cultivation method as well as cultivar and harvest time [49]. Many studies indicate that organic farming systems have a significant impact on the quality of strawberries produced, such as the use of organic and biological fertilization [50,51].

The fertilization treatments significantly influenced the number of petioles rhubarb per plant, their average weight per plant, the average weight per plant, and the yield dynamics (Table 6).

The number of petioles per plant ranged from 9.2^2^, in the case of the control plant, to 11.14, in the case of Ch.

The average weight of the petioles per plant was not significantly influenced by fertilization. However, this ranged from 39.11 g·plant^−1^, in the case of the control plant, to 42.92 g·plant^−1^, in the case of Mo.

The total yield ranged from 47.27 t·ha^−1^, in the case of the unfermented treatment, to 59.16 t·ha^−1^, in the case of Ch, although the differences between treatments were not significant.

The influence of different types of fertilizers on production indicators has also been demonstrated in vegetable species such as *Cynara cardunculus* L. [52,53].

The total quantity of stalks ranged from 1199.75 kg·ha^−1^, in the case of R_4_, up to 8044.75 kg·ha^−1^, in the case of the first harvesting season (R_1_) (Figure 1). The highest yields were recorded in the first three harvests and in the last four, whereas the lowest values were recorded in the first three eras, and respectively in the last four. The lowest values were recorded in the middle of the harvesting period (R_4_, R_5_, and R_6_).

Mouna et al. conducted a study on *Salvia officinalis* L., where phenophases led to higher production at the beginning of the harvesting period [54].

Fertilizer and harvesting times significantly influenced the dynamic yield of rhubarb petioles. Thus, the dynamic yield for the 40 experimental treatments varied widely from 560 kg·ha^−1^, in the case of the control at R_4_, up to 11,997 kg·ha^−1^, in the case of chemical fertilization (Ch) from the ninth harvest (R_9_). Highest values were recorded under Ch fertilization at R_7_ and that with Og from R_9_, which showed positive increases, compared to the control from the fourth harvest time (R_4_) (Table 7).

Corroborating the dynamics of the production with the climatic conditions, we can say that the production at the perennial plants has its peak at the beginning of the vegetation period when the average temperature increases above 5 °C and the soil has sufficient accumulated humidity from the winter season. The vegetative buds bloom in greater numbers, but they grow slower, they are more succulent, the petioles reach the normal size in a longer period, and they accumulate water in greater quantity, when the harvest is bigger. After harvesting and in more arid onditions, the plants grow slower; they recover in a longer period, because they start from dormant buds or those that appear in the respective year of culture.

The application of chemical fertilizers, which have high solubility, made the production from the second part of the vegetation period bigger, and between the control and the types of fertilization at the beginning of the year, there were not very big differences, because the first petioles initially grow from the accumulated reserves in rhizomes in the previous year.

## 3. Materials and Methods

### 3.1. Plant Material and Growth Conditions

A research study was carried out in 2017 at “V. Adamachi” Experimental Station of the Agronomic University of Iasi (47°19′25″N, 27°54′99″E, 150 m a.s.l.) on rhubarb (*Rheum rhabarbarum* L.) using root cuttings of the Victoria cv. The cambic chernozem soil is characterized by a medium fertility, with 3.1% organic matter, 32% of clay and pH = 6.6. During the experimental period, the average temperature was 18.54 °C, the precipitation was 293.5 mm, and the relative humidity of the air was 66.8% (Table 8).

The planting was practiced in 2012, with plants spaced 0.75 m along the rows, which were 1.00 m apart (density of 1.33 plants·m^−2^). Due to the dry agricultural year, three irrigations were practiced, each with 250 m^3^·ha^−1^, when the soil available water decreased below 80%. Flowering stems were removed 4 to 5 days after planting in order to favor the development of the edible part [55]. During the vegetation period, two manual hoeings were performed between rows and plants. No phytosanitary treatments were performed for protecting plants against diseases and pests. The biometric and biochemical determinations were made in 2017 on a rhubarb crop set up by seedlings to the fifth year of harvest.

### 3.2. Experimental Protocol

Three types of fertilization were compared in a split plot design experiment with three replications. Chemical (NPK = 16-16-16), Mycorrhizal-based formulate (Micoseeds MB^®^), and Organic (Orgevit^®^) treatments were compared to an unfertilized control (C). Chemical NPK 16-16-16 was produced by Ameropa Company^®^ (Romania), Micoseeds MB^®^ was from MsBiotech (Italy), and Orgevit^®^ was from MeMon BV (Netherlands). The biological product consists of an arbuscular mycorrhizal fungus (*Glomus* spp.), PGPR (*Pseudomonas* sp., *Bacillus* spp., *Streptomyces* sp.), and a fungus (*Trichoderma* sp.) in different proportions. Organic fertilizer is chicken manure formulate with pH 7, 4% N, 2.5% P2O5, 2.3% K2O, 1% MgO, 0.02% Fe, 0.01% Mn, 0.01% B, 0.01% Zn, 0.001% Cu, and 0.001% Mo. All fertilizers were supplied to the soil. Thus, the Ch fertilizer was applied in an amount of 425 kg ha^−1^. The product beneficial microorganisms-based Micoseed MB (Mo) was used in the dose of 60 kg ha^−1^. Og was applied at the dose of 2400 kg ha^−1^. Ch and Og fertilizers were supplied after the first petiole harvest (03.04) in five stages (04.04, 28.04, 12.05, 26.05, and 09.06); Micoseeds MB^®^ was applied in two stages, the first before planting (16.03) and the second one day after the first harvest (04.04) according to producer recommendations. For the determination of the doses of the fertilizers Og and Ch, the chemical composition of each product was taken into account, and for the Og, it was considered that 70% of the active substance of the product is assimilated in the first year after application.

Biometric and chemical determinations were performed at 10 harvest times: R_1_: 03.04, R_2_: 11.04, R_3_: 19.04, R_4_: 03.05, R_5_: 10.05, R_6_: 22.05, R_7_: 07.06, R_8_: 23.06, R_9_: 19.07, and R_10_: 12.08.

### 3.3. Biometric Measurements

From the field of research were collected stalks from each plant from the four variants taken in the experiment. For each variant, we determined the average quantities obtained by measuring the weight with the electronic scale. Petioles with a minimum diameter of 10–12 mm and a length of 25–30 cm were harvested. In the laboratory, weighing was performed on the analytical balance to determine the average weight of petioles per plant. To determine the length of the stalks, the measurements were made with the graduated ruler, with the unit of measure in centimeters, and the thickness of the petioles was determined with the electronic chisel (mm).

### 3.4. Samples Preparation

In order to prepare the material for the laboratory analyses, 10 stalks from each repetition were selected randomly for a total of 40 pieces. The petioles were cut into 1 cm fragments for drying under normal weather conditions. The drying was carried out on a Sanyo oven, type MOV-112F, at a temperature of 70 °C up to the total loss of water from the dry material in order to determine the quantity of water and the dry substance. The samples were crushed into small fragments of 0.1–1 mm. The last step consisted in the packaging and labeling of variants in order to carry out laboratory analyses to determine the polyphenol content and antioxidant capacity in plants.

### 3.5. HPLC-MS Analysis of Phenolic Compounds

In this application, the presence and content of different phenolic compounds in 70% ethanolic extracts were studied using an HPLC-MS, method which allows the simultaneous detection of several phenolic compounds with a single column pass [56].

The identification and quantification of polyphenolic compounds was carried out using an Agilent Technologies 1100 HPLC Series system (Agilent, Santa Clara, CA, USA) equipped with a G13311A binary gradient pump, G1322A degasser, column thermostat, G1316A UV detector, and G1313A autosampler. The HPLC system was coupled with an Agilent 1100 mass spectrometer (LC/MSD Ion Trap SL). For the separation, a reverse-phase analytical column was employed (Zorbax SB-C18 100 × 3.0 mm i.d., 3.5 μm particle) and the work temperature was set at 48 °C. The detection of the compounds was performed in both UV and MS mode [57,58,59]. The UV detector was set at 330 nm until 17.5 min, and then at 370 nm. The MS system was operated using an electrospray ion source in the negative mode. ChemStation and DataAnalysis software from Agilent were used for processing the chromatographic data. The mobile phase was a binary gradient: methanol and acetic acid 0.1% (*v*/*v*). The elution started with a linear gradient, beginning with 5% methanol and ending at 42% methanol, for 35 min; then, it had 42% methanol for the next 3 min. The flow rate was 1 mL·min^−1^, and the injection volume was 5 μL [60,61].

### 3.6. Antioxidant Activity Test

The rhubarb stalks’ antioxidant capacity was assessed in terms of radical scavenging activity, following the procedure described by Re et al. [62] with slight modifications [63]. In this respect, 500 μL aliquot of the extract or Trolox standard was added to 1 mL of DPPH methanol solution (74 mg·L^−1^). A daily prepared solution of 2,2-diphenyl-1-picryl-hydrazylhydrate (DPPH) showed a final absorption at 520 nm of 1.8 AU. The mixture was shaken and allowed to stand for 1 h at room temperature; then, the absorption was measured at 520 nm in a Lambda 25 spectrophotometer. The antiradical activity of the sample is inversely correlated with its purple color intensity. Aqueous solutions of Trolox at various concentrations were used for calibration (0.15–1.15 mmol·L^−1^). The results were expressed as μmol equivalents of Trolox (an analog of vitamin E) per g of sample (TEAC-Trolox Equivalent Antioxidant Capacity) [64].

### 3.7. Statistical Analysis

Data statistical processing was carried out by one-way and two-way ANOVA, and mean separations were performed through Tukey’s test using a SPSS version 21, referring to *p* ≤ 0.05 probability level.

## 4. Conclusions

The present study confirms that fertilizer management method may have a significant impact on the phenolic composition of rhubarb. The fact that there are qualitative variations between polyphenols at the same harvest date, under same environmental conditions, on the same cultivar indicates that they are influenced by the treatment used.

However, in addition, harvesting time under climatic conditions has an important effect on the quality and quantity of bioactive compounds.

The research shows that the biological fertilizer had a positive effect on the total phenol content, directly influencing the antioxidant capacity from rhubarb, compared to the chemical and organic fertilizers, which are very important in the human diet.

Biological fertilization provides satisfactory nutritional conditions for the accumulation of ferulic acid, p-coumaric acid, isoquercitrin, and quercetrol compared with rutoside, which accumulates in the highest quantities under organic fertilization. Polyphenolic compounds accumulate in petiols of rhubarb as follows: rutozide > isoquercitrin > ferulic acid > quercetrol > p-coumaric acid.

Chemical fertilization provided the highest yields, regardless of the harvesting period, but the differences with respect to organic and biological treatments were not, which means that there are favorable conditions for using the two fertilizers in sustainable agriculture, without restriction.

The treatments carried out and the harvesting period in accordance with the climatic conditions emphasize that there are the premises of promoting organic and biological treatments for a type of non-polluting, sustainable agriculture.

## Figures and Tables

**Figure 1 plants-09-00355-f001:**
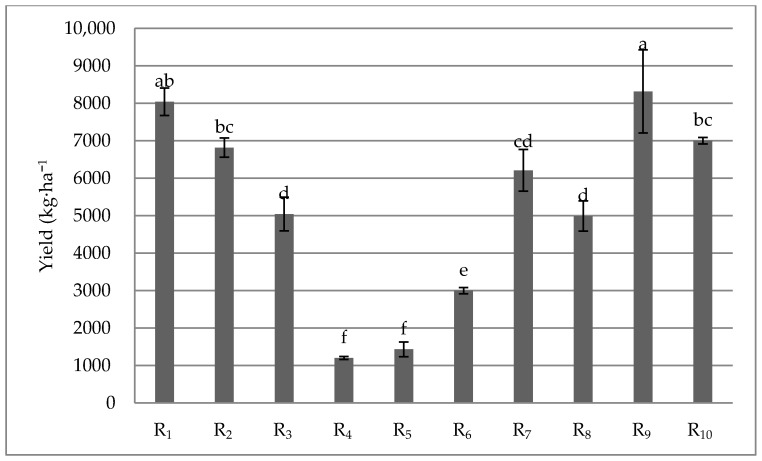
Yield dynamics influenced by harvest time (n = 3). R_1_–R_10_ (first harvest time—the 10th harvesting time). Within each column, the mean ± standard deviation of each variable is reported in correspondence with each experimental treatment. Along each line, values followed by different letters are significantly different according to Tukey’s test at *p* ≤ 0.05.

**Table 1 plants-09-00355-t001:** Main effects of the experimental factors on the total polyphenol content and antioxidant activity (n = 3).

Treatment	Total Phenols Content—TPC (mg GAE·g^−1^ d.w.)	Antioxidant Activity—AC (mmol Trolox·g^−1^ d.w.)
Fertilization		
Ch	1320.62 ± 111.71 ^b^	478.48 ± 41.43 ^c^
Mo	1605.03 ± 154.45 ^a^	877.07 ± 86.31 ^a^
Og	1477.95 ± 102.31 ^ab^	728.05 ± 50.68 ^b^
C	992.74 ± 87.35 ^c^	313.17 ± 26.78 ^d^
Harvest time		
R_1_	1345.63 ± 62.52 ^c^	582.73 ± 51.14
R_2_	1364.14 ± 47.86 ^c^	589.75 ± 48.89 ^c^
R_3_	1315.55 ± 80.32 ^cd^	565.12 ± 48.75 ^c^
R_4_	2450.83 ± 287.23 ^a^	1079.76 ± 165.65 ^a^
R_5_	1819.34 ± 213.24 ^b^	834.44 ± 147.00 ^b^
R_6_	1259.78 ± 240.31 ^cd^	586.83 ± 147.35 ^c^
R_7_	1164.37 ± 144.00 ^cd^	515.47 ± 69.14 ^c^
R_8_	1015.14 ± 109.14 ^d^	467.39 ± 73.83 ^c^
R_9_	1085.13 ± 77.13 ^cd^	477.34 ± 52.5 ^c^
R_10_	670.92 ± 50.3 ^e^	293.09 ± 31.19 ^d^

Ch—Chemical; Mo—Biological; Og—Organic; C—Control; R_1_–R_10_ (first harvest time—the 10th harvesting time); GAE—Gallic Acid Equivalents. Within each column; Mean ± standard deviation of each variable is reported in correspondence with each experimental treatment. Along each line, values followed by different letters are significantly different according to Tukey’s test at *p* ≤ 0.05.

**Table 2 plants-09-00355-t002:** Interaction between harvest time and fertilization on TPC (mg GAE·g^−1^ d.w.) (n = 3).

Harvest Time	Treatment			
	Ch	Mo	Og	C
R_1_	1431.34 ± 119.09 ^ab^	1427.73 ± 125.68 ^cd^	1332.73 ± 104.3 ^bc^	1190.73 ± 161.28 ^b^
R_2_	1449.88 ± 81.00 ^ab^	1450.65 ± 122.27 ^cd^	1324.53 ± 90.65 ^bc^	1231.48 ± 60.55 ^b^
R_3_	1610.11 ± 186.78 ^a^	1329.76 ± 161.31 ^cd^	1229.71 ± 69.75 ^bcd^	1092.62 ± 64.38 ^bc^
R_4_	1738.26 ± 144.62 ^a^	1915.12 ± 168.59 ^bc^	3966.56 ± 310.41 ^a^	2183.38 ± 295.73 ^a^
R_5_	1521.36 ± 85.00 ^a^	2954.52 ± 249.02 ^a^	1627.58 ± 111.39 ^b^	1173.89 ± 57.72 ^b^
R_6_	901.42 ± 104.57 ^bc^	2569.88 ± 311.75 ^b^	687.72 ± 39.01 ^d^	880.10 ± 51.86 ^bcd^
R_7_	1704.53 ± 141.82 ^a^	1285.07 ± 113.13	1218.81 ± 95.38 ^bcd^	449.08 ± 60.82 ^d^
R_8_	829.51 ± 46.34 ^c^	1364.39 ± 115.00 ^cd^	1324.28 ± 90.63 ^bc^	542.37 ± 26.67 ^cd^
R_9_	1188.14 ± 137.83 ^abc^	1092.31 ± 132.51 ^cd^	1310.19 ± 74.32 ^bcd^	749.87 ± 44.18 ^bcd^
R_10_	831.59 ± 70.09 ^c^	660.88 ± 45.23 ^d^	757.35 ± 37.24 ^cd^	433.86 ± 50.33 ^d^

Ch—Chemical; Mo—Biological; Og—Organic; C—Control; R_1_–R_10_ (first harvest time—the 10th harvesting time). Within each column; Mean ± standard deviation of each variable is reported in correspondence with each experimental treatment. Along each line, values followed by different letters are significantly different according to Tukey’s test at *p* ≤ 0.05.

**Table 3 plants-09-00355-t003:** Interaction between harvest time and fertilization on antioxidant capacity (mmol Trolox·g^−1^ d.w.) (n = 3).

Harvest Time	Treatment			
	Ch	Mo	Og	C
R_1_	518.60 ± 43.15 ^bc^	780.18 ± 68.68 ^cd^	656.52 ± 51.38 ^bc^	375.62 ± 50.88 ^b^
R_2_	525.32 ± 29.35 ^b^	792.70 ± 66.81 ^cd^	652.48 ± 44.66 ^bc^	388.48 ± 19.10 ^b^
R_3_	583.37 ± 67.67 ^a^	726.64 ± 88.15 ^cd^	605.77 ± 34.36 ^bcd^	344.68 ± 20.31 ^bc^
R_4_	629.80 ± 52.40 ^a^	1046.51 ± 92.13 ^bc^	1953.97 ± 152.91 ^a^	688.76 ± 93.29 ^a^
R_5_	551.22 ± 30.80 ^a^	1614.49 ± 136.08 ^a^	801.76 ± 54.87 ^b^	370.31 ± 18.21 ^b^
R_6_	326.60 ± 37.89 ^bc^	1404.31 ± 170.36 ^ab^	338.78 ± 19.22 ^d^	277.63 ± 16.36 ^bcd^
R_7_	617.58 ± 51.38 ^a^	702.22 ± 61.82 ^cd^	600.40 ± 46.99 ^bcd^	141.67 ± 19.19 ^d^
R_8_	300.55 ± 16.79 ^c^	745.57 ± 62.84 ^cd^	652.35 ± 44.65 ^bc^	171.09 ± 8.41 ^cd^
R_9_	430.49 ± 49.94 ^bc^	596.89 ± 72.41 ^cd^	645.41 ± 36.61 ^bcd^	236.55 ± 13.94 ^bcd^
R_10_	301.30 ± 34.95 ^c^	361.14 ± 43.81 ^d^	373.08 ± 21.16 ^cd^	136.86 ± 8.06 ^d^

Ch—Chemical; Mo—Biological; Og—Organic; C—Control; R_1_–R_10_ (first harvest time—the 10th harvesting time). Within each column, the mean ± standard deviation of each variable is reported in correspondence with each experimental treatment. Along each line, values followed by different letters are significantly different according to Tukey’s test at *p* ≤ 0.05.

**Table 4 plants-09-00355-t004:** Main effects of the experimental factors on the polyphenol compounds (n = 3).

Treatment	P-Coumaric Acid (µg·g^−1^)	Ferulic Acid (µg·g^−1^)	Isoquercitrin (µg·g^−1^)	Rutozid (µg·g^−1^)	Quercetrol (µg·g^−1^)
Fertilization					
Ch	9.86 ± 0.87 ^c^	21.67 ± 1.76 ^c^	37.04 ± 3.44 ^c^	473.2 ± 37.97 ^a^	16.55 ± 1.28 ^b^
Mo	15.46 ± 1.37 ^a^	31.45 ± 2.98 ^a^	78.65 ± 5.58 ^a^	479.08 ± 39.48 ^a^	20.20 ± 1.87 ^a^
Og	11.03 ± 0.84 ^bc^	22.23 ± 1.71 ^c^	61.82 ± 4.30 ^b^	490.97 ± 42.42 ^a^	15.94 ± 1.41 ^b^
C	11.58 ± 0.98 ^b^	27.06 ± 2.51 ^b^	68.2 ± 6.22 ^b^	372.57 ± 37.89 ^b^	11.17 ± 0.96 ^c^
Harvest time					
R_1_	6.43 ± 0.41 ^e^	16.46 ± 0.68 ^d^	47.26 ± 6.48 ^b^	491.69 ± 21.68 ^cd^	11.22 ± 0.78 ^c^
R_2_	6.72 ± 0.36 ^e^	16.76 ± 0.81 ^d^	48.66 ± 6.39 ^b^	497.92 ± 19.07 ^c^	11.38 ± 0.70 ^c^
R_3_	9.12 ± 1.11 ^cde^	25.55 ± 3.89 ^bc^	50.93 ± 6.58 ^b^	462.48 ± 31.60 ^cd^	11.73 ± 0.69 ^bc^
R_4_	18.45 ± 2.52 ^b^	24.53 ± 1.88 ^c^	75.58 ± 9.41 ^a^	904.12 ± 110.43 ^a^	21.09 ± 1.20 ^a^
R_5_	8.22 ± 0.92 ^de^	15.69 ± 0.88 ^d^	56.32 ± 6.51 ^b^	666.61 ± 65.75 ^b^	12.28 ± 1.13 ^bc^
R_6_	10.32 ± 1.32 ^cd^	31.37 ± 1.82 ^b^	84.83 ± 16.41 ^a^	382.32 ± 76.61 ^de^	15.31 ± 1.76 ^b^
R_7_	23.57 ± 4.04 ^a^	38.44 ± 3.56 ^a^	58.63 ± 8.76 ^b^	344.47 ± 54.88 ^e^	19.71 ± 3.00 ^a^
R_8_	7.61 ± 0.95 ^de^	39.96 ± 3.20 ^a^	79.44 ± 11.55 ^a^	281.38 ± 30.47 ^ef^	13.94 ± 1.61 ^bc^
R_9_	17.25 ± 1.55 ^b^	26.30 ± 1.53 ^bc^	80.21 ± 7.03 ^a^	312.55 ± 20.48 ^e^	21.64 ± 1.51 ^a^
R_10_	12.13 ± 1.24 ^c^	20.99 ± 0.93 ^cd^	32.43 ± 1.36 ^c^	196.02 ± 16.67 ^f^	21.37 ± 2.63 ^a^

Ch—Chemical; Mo—Biological; Og—Organic; C—Control; R_1_–R_10_ (first harvest time—the 10th harvesting time). Within each column, mean ± standard deviation of each variable is reported in correspondence with each experimental treatment. Along each line, values followed by different letters are significantly different according to Tukey’s test at *p* ≤ 0.05.

**Table 5 plants-09-00355-t005:** Interaction between fertilization type and harvest on the polyphenols composition (n = 3).

Treatment	P-Coumaric Acid (µg·g^−1^)	Ferulic Acid (µg·g^−1^)	Isoquercitrin (µg·g^−1^)	Rutozid (µg·g^−1^)	Quercetrol (µg·g^−1^)
Interaction of factors					
Ch × R_1_	5.84 ± 0.49 ^ns^	14.02 ± 0.78 ^ns^	26.12 ± 2.17 ^b^	547.04 ± 42.81 ^ns^	12.12 ± 1.64 ^ab^
Mo × R_1_	8.22 ± 0.73 ^ns^	18.50 ± 1.56 ^ns^	62.36 ± 5.49 ^a^	454.58 ± 61.57 ^ns^	12.16 ± 0.68 ^ab^
Og × R_1_	5.96 ± 0.47 ^ns^	16.44 ± 1.12 ^ns^	29.22 ± 2.29 ^b^	478.24 ± 26.72 ^ns^	12.98 ± 1.09 ^a^
C × R_1_	5.68 ± 0.77 ^ns^	16.86 ± 0.83 ^ns^	71.34 ± 9.66 ^a^	486.90 ± 41.04 ^ns^	7.64 ± 0.52 ^b^
Ch × R_2_	6.58 ± 0.37 ^ab^	14.52 ± 1.69 ^ns^	26.28 ± 1.47 ^b^	553.40 ± 37.88 ^ns^	12.20 ± 0.60 ^a^
Mo × R_2_	8.40 ± 0.71 ^a^	18.80 ± 2.28 ^ns^	64.02 ± 5.40 ^a^	461.20 ± 22.68 ^ns^	12.32 ± 1.02 ^a^
Og × R_2_	6.06 ± 0.42 ^b^	16.68 ± 0.95 ^ns^	30.54 ± 2.09 ^b^	473.26 ± 39.38 ^ns^	12.96 ± 1.14 ^a^
C × R_2_	5.86 ± 0.29 ^b^	17.02 ± 1.00 ^ns^	73.80 ± 3.63 ^a^	503.84 ± 44.35 ^ns^	8.04 ± 0.63 b
Ch × R_3_	7.62 ± 0.43 ^bc^	24.28 ± 1.90 ^b^	28.58 ± 3.32 ^b^	607.96 ± 47.58 ^a^	12.28 ± 1.66 ^ns^
Mo × R_3_	14.84 ± 0.87 ^a^	45.52 ± 6.17 ^a^	68.66 ± 5.37 ^a^	376.38 ± 50.98 ^b^	12.28 ± 0.69 ^ns^
Og × R_3_	8.82 ± 0.78 ^b^	17.20 ± 0.96 ^b^	31.66 ± 4.29 ^b^	429.82 ± 24.01 ^ab^	13.38 ± 1.13 ^ns^
C × R_3_	5.20 ± 0.41 ^c^	15.18 ± 1.28 ^b^	74.82 ± 4.18 ^a^	435.76 ± 36.73 ^ab^	8.98 ± 0.61 ^ns^
Ch × R_4_	12.42 ± 1.68 ^bc^	19.22 ± 1.32 ^b^	37.82 ± 3.19 ^c^	646.56 ± 44.25 ^b^	18.88 ± 0.93 ^ns^
Mo × R_4_	22.06 ± 1.23 ^ab^	33.38 ± 1.64 ^a^	65.56 ± 4.49 ^b^	599.06 ± 29.45 ^b^	25.50 ± 2.96 ^ns^
Og × R_4_	10.02 ± 0.78 ^c^	21.26 ± 2.47 ^b^	121.04 ± 5.95 ^a^	1442.24 ± 167.31 ^a^	21.08 ± 2.56 ^ns^
C × R_4_	29.28 ± 3.97 ^a^	24.28 ± 2.95 ^ab^	77.90 ± 9.04 ^b^	928.62 ± 112.65 ^b^	18.88 ± 1.07 ^ns^
Ch × R_5_	8.82 ± 0.49 ^ab^	12.16 ± 0.69 ^b^	31.66 ± 3.84 ^c^	578.28 ± 32.80 ^b^	12.28 ± 0.72 ^b^
Mo × R_5_	12.42 ± 1.68 ^a^	15.18 ± 1.26 ^ab^	87.14 ± 4.94 ^a^	1017.68 ± 69.65 ^a^	17.78 ± 1.57 ^a^
Og × R_5_	6.42 ± 0.36 ^b^	18.22 ± 1.61 ^a^	44.00 ± 2.59 ^c^	584.22 ± 28.72 ^b^	10.08 ± 0.79 ^b^
C × R_5_	5.20 ± 0.44 ^b^	17.20 ± 1.34 ^ab^	62.48 ± 3.49 ^b^	486.24 ± 56.41 ^b^	8.98 ± 1.21 ^b^
Ch × R_6_	5.20 ± 0.46 ^b^	24.28 ± 3.29 ^b^	40.92 ± 3.45 ^b^	296.22 ± 35.93 ^b^	14.48 ± 1.21 ^b^
Mo × R_6_	7.62 ± 0.60 ^b^	37.44 ± 2.09 ^a^	124.12 ± 8.50 ^a^	806.88 ± 67.13 ^a^	24.40 ± 2.15 ^a^
Og × R_6_	14.84 ± 2.01 ^a^	30.36 ± 2.56 ^ab^	25.50 ± 1.25 ^b^	198.24 ± 17.45 ^b^	11.18 ± 0.88 ^b^
C × R_6_	13.64 ± 1.2 ^a^	33.38 ± 2.28 ^ab^	148.78 ± 17.26 ^a^	227.94 ± 17.84 ^b^	11.18 ± 1.51 ^b^
Ch × R_7_	17.24 ± 0.96 ^b^	30.36 ± 1.49 ^b^	37.82 ± 4.59 ^bc^	602.02 ± 81.54 ^a^	33.20 ± 1.85 ^a^
Mo × R_7_	46.14 ± 3.89 ^a^	51.58 ± 5.99 ^a^	56.32 ± 3.20 ^b^	322.94 ± 18.04 ^b^	23.30 ± 1.96 ^b^
Og × R_7_	16.04 ± 1.10 ^b^	27.32 ± 2.14 ^b^	105.64 ± 6.22 ^a^	331.86 ± 27.97 ^b^	15.58 ± 1.06 ^c^
C × R_7_	14.84 ± 0.73 ^b^	44.50 ± 6.03 ^ab^	34.74 ± 2.72 ^c^	121.06 ± 8.29 ^c^	6.78 ± 0.34 ^d^
Ch × R_8_	4.00 ± 0.22 ^b^	27.32 ± 2.27 ^b^	31.66 ± 4.29 ^c^	275.44 ± 13.54 ^ab^	12.28 ± 1.43 ^b^
Mo × R_8_	5.20 ± 0.44 ^b^	47.54 ± 4.18 ^a^	130.30 ± 7.28 ^a^	325.92 ± 37.81 ^a^	22.20 ± 2.69 ^a^
Og × R_8_	11.22 ± 0.77 ^a^	36.42 ± 2.85 ^ab^	96.38 ± 8.12 ^b^	385.30 ± 46.74 ^a^	10.08 ± 0.57 ^b^
C × R_8_	10.02 ± 0.49 ^a^	48.56 ± 6.58 ^a^	59.40 ± 4.07 ^c^	138.86 ± 7.88 ^b^	11.18 ± 0.66 ^b^
Ch × R_9_	24.46 ± 2.84 ^a^	28.32 ± 1.58 ^a^	71.74 ± 5.97 ^ab^	355.60 ± 20.95 ^ns^	22.20 ± 1.74 ^ns^
Mo × R_9_	17.24 ± 2.09 ^ab^	26.30 ± 2.22 ^ab^	96.38 ± 8.48 ^a^	257.62 ± 22.68 ^ns^	27.70 ± 3.75 ^ns^
Og × R_9_	13.64 ± 0.78 ^b^	19.22 ± 1.32 ^b^	102.56 ± 8.03 ^a^	379.36 ± 29.69 ^ns^	18.88 ± 1.06 ^ns^
C × R_9_	13.64 ± 0.80 ^b^	31.36 ± 1.54 ^a^	50.16 ± 6.79 ^b^	257.62 ± 34.89 ^ns^	17.78 ± 1.50 ^ns^
Ch × R_10_	6.42 ± 0.75 ^c^	22.26 ± 2.58 ^ns^	37.82 ± 2.11 ^ns^	269.50 ± 22.42 ^a^	15.58 ± 1.06 ^bc^
Mo × R_10_	12.42 ± 1.51 ^b^	20.24 ± 2.46 ^ns^	31.66 ± 2.67 ^ns^	168.56 ± 14.84 ^b^	24.40 ± 1.20 ^ab^
Og × R_10_	17.24 ± 0.98 ^a^	19.22 ± 1.09 ^ns^	31.66 ± 2.17 ^ns^	207.16 ± 16.21 ^ab^	33.20 ± 3.85 ^a^
C × R_10_	12.42 ± 0.73 ^b^	22.26 ± 1.31 ^ns^	28.58 ± 1.41 ^ns^	138.86 ± 18.81 ^b^	12.28 ± 1.49 ^c^

Ch—Chemical; Mo—Biological; Og—Organic; C—Control; R_1_–R_10_ (first harvest time—the 10th harvesting time). Within each column, mean ± standard deviation of each variable is reported in correspondence with each experimental treatment. Along each line, values followed by different letters are significantly different according to Tukey’s test at *p* ≤ 0.05.

**Table 6 plants-09-00355-t006:** Influence of fertilization on some production indicators (n = 3).

Fertilization	Average Number of Stalks per Plant	Average Weight per Plant (g)	Average Weight of Stalks per Plant (g)	Yield (t·ha^−1^)
	ns	ns	ns	ns
Ch	11.14 ± 0.53	443.8 ± 10.77	41.07 ± 1.43	59.16 ± 5.21
Mo	9.32 ± 0.32	368.80 ± 13.82	42.92 ± 1.42	49.16 ± 3.85
Og	10.22 ± 0.06	395.20 ± 19.14	40.31 ± 1.17	52.68 ± 7.13
C	9.22 ± 0.47	354.60 ± 6.86	39.11 ± 1.05	47.27 ± 2.64

Ch—Chemical; Mo—Biological; Og—Organic; C—Control. Within each column, the mean ± standard deviation of each variable is reported in correspondence with each experimental treatment. Along each line, values followed by different letters are significantly different according to Tukey’s test at *p* ≤ 0.05, ns—no statistically significant difference.

**Table 7 plants-09-00355-t007:** Interaction between harvest time and fertilization on dynamic yield (kg·ha^−1^) (n = 3).

Harvest Time	Treatment			
	Ch	Mo	Og	C
R_1_	7278 ± 498.1 ^b–g^	7838 ± 385.37 ^b–e^	8665 ± 1005.19 ^bc^	8398 ± 1018.76 ^bd^
R_2_	5865 ± 332.67 ^c–i^	5865 ± 487.97 ^c–i^	7438 ± 654.77 ^b–f^	8105 ± 634.26 ^b–e^
R_3_	5092 ± 689.69 ^e–l^	5519 ± 308.34 ^c–j^	4212 ± 355.01 ^g–n^	5359 ± 366.77 ^d–k^
R_4_	800 ± 39.33 ^o–p^	2106 ± 244.31 ^l–p^	1333 ± 161.71 ^n–p^	560 ± 31.76 ^p^
R_5_	1013 ± 79.27 ^o–p^	2266 ± 306.92 ^k–p^	1600 ± 89.39 ^m–p^	853 ± 71.89 ^op^
R_6_	3066 ± 209.84 ^i–p^	2453 ± 120.61 ^j–p^	3946 ± 328.31 ^h–o^	2533 ± 222.98 ^j–p^
R_7_	11864 ± 928.43 ^a^	2853 ± 386.42 ^i–p^	5199 ± 290.46 ^e–l^	4932 ± 415.7 ^e–l^
R_8_	6612 ± 452.52 ^b–h^	5305 ± 260.83 ^d–k^	2800 ± 324.82 ^i–p^	5252 ± 637.12 ^d–l^
R_9_	11997 ± 1624.93 ^a^	6985 ± 390.24 ^b–h^	9758 ± 822.45 ^ab^	4532 ± 310.17 ^f–m^
R_10_	5572 ± 273.96 ^c–j^	7971 ± 924.68 ^b–e^	7731 ± 937.84 ^b–e^	6745 ± 382.59 ^b–h^

Ch—Chemical; Mo—Biologic; Og—Organic; C—Control; R_1_–R_10_ (first harvest time—the 10th harvesting time). Within each column, the mean ± standard deviation of each variable is reported in correspondence with each experimental treatment. Along each line, values followed by different letters are significantly different according to Tukey’s test at *p* ≤ 0.05.

**Table 8 plants-09-00355-t008:** Temperature, rainfall, relative humidity, and sunlight during the experiment.

Month/Year	Temperature Average (°C)	Total Rainfall (mm)	Relative Humidity (%)	Sunlight (hours)
April/2017	10.3	89.1	67	205.5
May/2017	16.5	71.0	68	278.4
June/2017	21.7	46.6	65	293.4
July/2017	22.0	47.8	68	291.0
August/2017	22.2	39.0	66	290.5

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
