# Peer review of "Dynamic of Phenolic Compounds, Antioxidant Activity, and Yield of Rhubarb under Chemical, Organic and Biological Fertilization"

_plants, 2020, doi:10.3390/plants9030355_

Round 1
Reviewer 1 Report
Generally, the manuscript is novel. However, there are few formatting and organization of the data that need to be considered. I suggest that some of the tables be converted to graphs. Also, the number of replicates that gave the means need to be presented beneath the tables.
If these are fixed, the manuscript could be considered for publication.
Author Response
Dear Editor,
we have addressed the Reviewer 1& 2 recommendations, highlighting the modifications/amendments in red colour.
We wish to thank the Reviewers for his beneficial contribution aimed to improve our manuscript.
Reviewer 1
Generally, the manuscript is novel. However, there are few formatting and organization of the data that need to be considered. I suggest that some of the tables be converted to graphs. Also, the number of replicates that gave the means need to be presented beneath the tables.
Answer: we have addressed the above recommendation
Regarding to the number of replicates, for each version number of replicates were 3. For each table was presented number of replicates, like (n±3).
Because in the tables are more data presented, I preferred to use tables, it is much easier to reading. Converting tables into graphics will be increase page number of manuscript and also I am not sure being a breezy graph.
There have been additions to the text and bibliography.
The bibliography was rearranged and cited.

Reviewer 2 Report
The authors investigated the chemical composition of rhubarb petioles depending on the fertilizers used. Authors stated that “for the determination of the doses of the fertilizers Og and Ch the chemical composition of each product was taken into account”. Please, provide the results of chemical analyses and the formulas on which the fertilizers’ doses were based on. It is crucial for understanding the idea of experiments and mechanisms affecting plants' reactions to a particular type of fertilizer.
Regarding the experimental treatments, the fertilizers have a completely different composition and biological action. I get the feeling that the Authors try to compare incomparably. It is clearly seen in the discussion. The discussion is a simple comparison of the numerical data with the data of other authors. And there is no explanation of the reasons for significant differences observed. For example, why microorganisms fertilizer affect increased polyphenol content and antioxidant activity? What kind of stress could be involved in this reaction? Etc.
The authors performed sequential analyses of rhubarb petioles chemical composition. It but the main reason could be meteorological conditions, not experimental treatments. I suspect that Authors planed such an experiment considering weather impact, but it should be explained in the manuscript. I think that the conclusions on the best harvest period basing on the one season are speculative.
To sum up:
the main weakness: comparison of fertilizers of different compositions and action (the way to improve could be the providing the information on fertilizer types and theirs action, one-year field experiment, lack of deep discussion (could be improved). A lot of data - a few weak conclusions.
Strengths: complex and sequential analyses. Generally well-written text.
Author Response
Dear Editor,
we have addressed the Reviewer 1& 2 recommendations, highlighting the modifications/amendments in red colour.
We wish to thank the Reviewers for his beneficial contribution aimed to improve our manuscript.
Reviewer 2
The authors investigated the chemical composition of rhubarb petioles depending on the fertilizers used. Authors stated that “for the determination of the doses of the fertilizers Og and Ch the chemical composition of each product was taken into account”. Please, provide the results of chemical analyses and the formulas on which the fertilizers’ doses were based on. It is crucial for understanding the idea of experiments and mechanisms affecting plants' reactions to a particular type of fertilizer.
Answer: we have addressed the above recommendation included fertilization composition in manuscript
Regarding the experimental treatments, the fertilizers have a completely different composition and biological action. I get the feeling that the Authors try to compare incomparably. It is clearly seen in the discussion. The discussion is a simple comparison of the numerical data with the data of other authors. And there is no explanation of the reasons for significant differences observed. For example, why microorganisms fertilizer affect increased polyphenol content and antioxidant activity? What kind of stress could be involved in this reaction?
Answer: we have addressed the above recommendation.
The role of AMF should be taken into consideration in the Mo treatment as an abiotic stress according to environmental conditions (Celebi et al., 2010).
Higher polyphenol content can be attributed to the effect of mycorrhizal associations, which induce a decrease in carbohydrate content in cells (Ferrol and Perez-Tienola, 2009).
Solfanelli et al., 2006 justify the higher content of polyphenolic compounds by the fact that in warmer periods, with less precipitation, the sugar content increases in plants and thus the activity of AMF is higher.
In general, polyphenols are secondary metabolism compounds that are produced by plants for defensive purposes when in states of increased biotic or abiotic stress. In perennial species, as in Populus sp., high TPC and AC are also accounted for by water stress and higher temperatures (Popovic et al.,2016 )
The authors performed sequential analyses of rhubarb petioles chemical composition. It but the main reason could be meteorological conditions, not experimental treatments. I suspect that Authors planed such an experiment considering weather impact, but it should be explained in the manuscript. I think that the conclusions on the best harvest period basing on the one season are speculative.
Corroborating the dynamics of the production with the climatic conditions, we can say that the production at the perennial plants has its peak at the beginning of the vegetation period when the average temperature increases above 50C and the soil has sufficient accumulated humidity from the winter season. The vegetative buds bloom in greater numbers, but they grow slower, they are more succulent, the petioles reach the normal size in a longer period, they accumulate water in greater quantity, when the harvest is bigger. After harvesting and in more arid conditions, the plants grow slower; they recover in a longer period, because they start from dormant buds or those that appear in the respective year of culture.
The application of chemical fertilizers, which have high solubility, makes the production from the second part of the vegetation period to be bigger, and between the control and the types of fertilization at the beginning of the year there are not very big differences, because the first petioles initially grow from the accumulated reserves in rhizomes, in the previous year.
To sum up:the main weakness: comparison of fertilizers of different compositions and action (the way to improve could be the providing the information on fertilizer types and theirs action, one-year field experiment, lack of deep discussion (could be improved). Strengths: complex and sequential analyses. Generally well-written text.
The content of biologically active compounds depends mainly on the cultivation method as well as cultivar and harvest time (Ponder and Hallmann, 2019). The idea of using fertilizers with different compositions also aims at the possibility of using them in different growing systems, such as sustainable agriculture or ecological.
Many studies indicate that organic farming systems have a significant impact on the quality of strawberries produced (Crecente-Campo et al., 2012) like are organic and biological fertilization.
A lot of data - a few weak conclusions.
Answer: we have addressed the above recommendation.
The bibliography was added, rearranged and cited.
- Celebi, S.Z.; Demir, S.; Cele bi, R.; Durak, E.D.; Yilmaz, I.H. The effect of Arbuscular Mycorrhizal Fungi (AMF) applications on the silage maize (Zea mays L.) yield in different irrigation regimes. Eur. J. Soil Biol. 2010, 46, 302-305. 10.1016/j.ejsobi.2010.06.002
- Ferrol N., Pérez-Tienda J. Coordinated Nutrient Exchange in Arbuscular Mycorrhiza in Mycorrhizas - Functional Processes and Ecological Impact; Azcón-Aguilar C., Barea J., Gianinazzi S., Gianinazzi-Pearson V.; Publisher: Springer Berlin Heidelberg, Germany, 2009, pp. 73-87.
- Solfanelli, C.; Poggi, A.; Loreti, E.; Alpi, A.; Perata, P. Sucrose-specific induction of the anthocyanin biosynthetic pathway in Arabidopsis. Plant Physiol. 2006, 140, 637-646. 10.1104/pp.105.072579
- Avio, L.; Sbrana, C.; Giovannetti, M.; Frassinetti, S. Arbuscular mycorrhizal fungi affect total phenolics content and antioxidant activity in leaves of oak leaf lettuce varieties. Sci. Hortic. 2017, 224, 265-271. 10.1016/j.scienta.2017.06.022
- Ponder, A.; Hallmann, E. The effects of organic and conventional farm management and harvest time on the polyphenol content in different raspberry cultivars. Food Chem. 2019, 301, 1-8. 10.1016/j.foodchem.2019.125295
- Crecente-Campo, J.; Nunes-Damaceno, M.; Romero-Rodriguez, M.A.; Vazquez-Oderiz, M.L. Color, anthocyanin pigment, ascorbic acid and total phenolic compound determination in organic versus conventional strawberries (Fragaria x ananassa Duch, cv Selva). J. Food Compos. Anal. 2012, 28, 23-30. 10.1016/j.jfca.2012.07.004
- Popovic, B.M.; Stajner, D.; Zdero-Pavlovic, R.; Tumbas-Saponjac, V.; Canadanovic-Brunet, J.; Orlovic, S. Water stress induces changes in polyphenol profile and antioxidant capacity in poplar plants (Populus spp.). Plant Physiol. Biochem. 2016, 105, 242-250. 10.1016/j.plaphy.2016.04.036
Kind regards,
Vasile Stoleru

Round 2
Reviewer 2 Report
The Authors sufficiently addressed my comments and the manuscript was significantly improved, so I recommend to accept.
The detailed comment: what does it mean (n±3) in Tables' headings? n - commonly means the sample size or the number of trials. It should be specified.
Author Response
Reviewer 1
The detailed comment: what does it mean (n±3) in Tables' headings? n - commonly means the sample size or the number of trials. It should be specified.
Answer: regarding to the comments I had change (n=3) - represent number of trials.
